# Characterizing Aptamer Interaction with the Oncolytic Virus VV-GMCSF-Lact

**DOI:** 10.3390/molecules29040848

**Published:** 2024-02-14

**Authors:** Maya A. Dymova, Daria O. Malysheva, Victoria K. Popova, Elena V. Dmitrienko, Anton V. Endutkin, Danil V. Drokov, Vladimir S. Mukhanov, Arina A. Byvakina, Galina V. Kochneva, Polina V. Artyushenko, Irina A. Shchugoreva, Anastasia V. Rogova, Felix N. Tomilin, Anna S. Kichkailo, Vladimir A. Richter, Elena V. Kuligina

**Affiliations:** 1Institute of Chemical Biology and Fundamental Medicine, Siberian Branch of the Russian Academy of Sciences, Lavrentiev av. 8, 630090 Novosibirsk, Russia; d.malysheva@g.nsu.ru (D.O.M.); fomenkoniboch@gmail.com (V.K.P.); elenad@niboch.nsc.ru (E.V.D.); aend@niboch.nsc.ru (A.V.E.); d.drokov@g.nsu.ru (D.V.D.); v.mukhanov@g.nsu.ru (V.S.M.); mytrilliangalaxy@gmail.com (A.A.B.); richter@niboch.nsc.ru (V.A.R.); kuligina@niboch.nsc.ru (E.V.K.); 2Department of Natural Sciences, Novosibirsk State University, Pirogova str. 1, 630090 Novosibirsk, Russia; 3State Research Center of Virology and Biotechnology “Vector”, 630559 Koltsovo, Russia; kochneva@vector.nsc.ru; 4Laboratory for Biomolecular and Medical Technologies, Krasnoyarsk State Medical University Named after Prof. V.F. Voyno-Yasenetsky, Partizana Zheleznyaka str. 1, 660022 Krasnoyarsk, Russia; art_polly@mail.ru (P.V.A.); shchugorevai@mail.ru (I.A.S.); arogova1927@gmail.com (A.V.R.); annazamay@yandex.ru (A.S.K.); 5Federal Research Center KSC SB RAS, 50 Akademgorodok, 660036 Krasnoyarsk, Russia; felixnt@gmail.com; 6Kirensky Institute of Physics, 50/38 Akademgorodok, 660012 Krasnoyarsk, Russia

**Keywords:** aptamer, oncolytic virus, glioma, dynamic light scattering, microscale thermophoresis

## Abstract

Aptamers are currently being investigated for their potential to improve virotherapy. They offer several advantages, including the ability to prevent the aggregation of viral particles, enhance target specificity, and protect against the neutralizing effects of antibodies. The purpose of this study was to comprehensively investigate an aptamer capable of enhancing virotherapy. This involved characterizing the previously selected aptamer for vaccinia virus (VACV), evaluating the aggregation and molecular interaction of the optimized aptamers with the recombinant oncolytic virus VV-GMCSF-Lact, and estimating their immunoshielding properties in the presence of human blood serum. We chose one optimized aptamer, NV14t_56, with the highest affinity to the virus from the pool of several truncated aptamers and built its 3D model. The NV14t_56 remained stable in human blood serum for 1 h and bound to VV-GMCSF-Lact in the micromolar range (Kd ≈ 0.35 μM). Based on dynamic light scattering data, it has been demonstrated that aptamers surround viral particles and inhibit aggregate formation. In the presence of serum, the hydrodynamic diameter (by intensity) of the aptamer–virus complex did not change. Microscale thermophoresis (MST) experiments showed that NV14t_56 binds with virus (EC50 = 1.487 × 10^9^ PFU/mL). The analysis of the amplitudes of MST curves reveals that the components of the serum bind to the aptamer–virus complex without disrupting it. In vitro experiments demonstrated the efficacy of VV-GMCSF-Lact in conjunction with the aptamer when exposed to human blood serum in the absence of neutralizing antibodies (Nabs). Thus, NV14t_56 has the ability to inhibit virus aggregation, allowing VV-GMCSF-Lact to maintain its effectiveness throughout the storage period and subsequent use. When employing aptamers as protective agents for oncolytic viruses, the presence of neutralizing antibodies should be taken into account.

## 1. Introduction

Grade III–IV gliomas are the most aggressive type of brain tumor and are characterized by high mortality, low life expectancy and high recurrence rates [1]. The standard treatment approaches, including surgical resection followed by temozolomide chemotherapy and radiation therapy, often fail to achieve patient recovery or significantly improve survival rates [2]. Thus, the development of novel treatment strategies and effective therapeutic agents remains a crucial biomedical challenge. The use of oncolytic virotherapy is a promising approach in the treatment of oncological diseases. This therapy demonstrated its potential through the targeted destruction of tumor cells, the ability to modulate the patient’s immune system and its compatibility with other treatment modalities such as chemotherapy and radiation therapy [3]. ICBFM SB RAS, in collaboration with the SRC VB “Vector”, has developed an anti-tumor drug based on the recombinant strain VV-GMCSF-Lact of the vaccinia virus [4]. The virus is currently undergoing clinical trials as a treatment for breast cancer (ClinicalTrials.gov identifier: NCT05376527). Additionally, we have demonstrated through in vitro and in vivo studies that VV-GMCSF-Lact exhibits strong oncolytic activity and antitumor efficacy against human glioma [5].

When using oncolytic viruses, it is important to consider that they can induce a robust immune response. The presence of antiviral immunity can hinder the effectiveness of virotherapy against tumors, as antibodies can neutralize the viral particles and impede their ability to reach the tumor. Therefore, the question arises regarding the delivery of the virus to the tumor site with repeated injections [6].

To address this issue, several approaches have been proposed, including (1) the initial selection of a viral vector for creating the oncolytic virus (enveloped viruses provide better protection against the complement system and neutralizing antibodies); (2) deliberate suppression of the immune system’s antiviral responses; (3) choosing appropriate methods for viral drug administration; (4) employing shielding cells or molecules (such as carrier cells, polyethylene glycol, CAR-T cells, chemical shells, chimeric viruses, aptamers, etc.) [7].

To date, numerous aptamers have been published for various viruses. These aptamers serve multiple purposes, such as diagnosis, therapy, and shielding against neutralizing antibodies. The last-mentioned application is particularly crucial in the field of oncolytic viruses [8,9,10,11]. The usage of the aptamers could additionally mitigate virus aggregation, increasing its infectivity and stability in human blood serum [12]. Viral vectors used in vaccination programs, as well as in genetic engineering for the development of anticancer drugs, require stabilization at low temperatures. Aptamers have been shown to increase viral infectivity by 1.4 log after 60 freeze–thaw cycles in a plaque formation assay and also prevent viral aggregation [13]. 

Earlier, a highly specific DNA aptamer (NV14) was selected against intact vaccinia virus using the SELEX technology [14]. This DNA aptamer originated from an 80 nt DNA library that contained a random 40 nt region (5′-CTC. CTC TGA CTG TAA CCA CG -N40- GC ATA GGT AGT CCA GAA GCC-3′). Through successive rounds of positive and negative selection, highly specific and selective aptamers were obtained, which do not bind to plasma proteins. NV14 was utilized to create an impedimetric aptasensor capable of distinguishing between viable and nonviable viruses. In this work, we aimed to carry out the molecular optimization of this aptamer using molecular-dynamic and quantum chemical modeling, and subsequently examine its binding to the oncolytic virus VV-GMCSF-Lact [13].

Various methods can be used to characterize the mechanisms of interaction between an aptamer and its molecular target. These methods include flow cytometry, which can be used to assess binding efficacy and estimate the dissociation constant [15]; dynamic light scattering (DLS), which can be utilized for evaluating particle size, aggregation, and electrostatic stability [16]; and microscale thermophoresis (MST), which allows for the measurement of the half-maximal effective concentration of a virus (EC50) [17]. In this study, we characterized the interaction of the aptamer with the oncolytic virus in the presence and absence of human blood serum, which contained neutralizing antibodies. Additionally, we demonstrated the cytotoxic effect of the virus in vitro in the presence of the aptamer and sera.

## 2. Results

### 2.1. Molecular Dynamic Simulations of Aptamer to Vaccinia Virus

For the molecular optimization, a previously discovered aptamer NV14 with good affinity and specificity for live vaccinia virus (VACV, Jennerex Inc., Ottawa, ON, Canada) was used [14]. Typically, during the SELEX (Systematic Evolution of Ligands by Exponential Enrichment) procedure, aptamers ranging from 80 to 100 nucleotides in length are selected, which may seem rather long for achieving selective binding. Truncation of these aptamers can enhance their specificity and affinity, reduce their cost, and simplify their synthesis.

Based on the nucleotide sequence, the secondary structure of the 80-nucleotide aptamer was modeled and then the corresponding 3D model was constructed (Figure 1). To achieve a structure that accurately reflects the conformation of aptamers in solution, molecular dynamics (MD) calculation was performed. The temperature and ion environment for the MD simulations were chosen to mimic the natural environment of the aptamer in vitro and in vivo. It was determined that 200 ns long MD simulations allow one to guarantee the structural stability of the aptamers, despite their high flexibility [18,19,20].

Based on the established secondary and spatial structure of the NV14 aptamer, its nucleotide sequence was truncated. Given the different spatial arrangements of complementary and single-stranded regions, aptamer modification was performed by truncating the nucleotides present in the primers. Aptamer NV14 was truncated by 11 nucleotides from the 5’ end and 12 nucleotides from the 3’ end (colored in dark blue, Figure 1). A secondary and three-dimensional structure model was created using a truncated nucleotide sequence containing 56 nucleotides. The modeling conditions used for the NV14t_56 aptamer were the same as those employed for the full-length aptamer. 

Similarly, we truncated all the aptamers proposed in [14]. After conducting a series of preliminary experiments using flow cytometry, as described in [21], we determined that NV14t_56 demonstrated the most efficient binding to our oncolytic virus VV-GMCSF-Lact. Consequently, we exclusively utilized this aptamer in subsequent experiments. Herein, we present a detailed description of its secondary and spatial structure.

### 2.2. Stability of Aptamer–Virus Complex in the Presence of Human Blood Serum and Aptamer Binding to VV-GMCSF-Lact

To assess the stability of the aptamer NV14t_56 in the presence of human blood serum containing neutralizing antibodies (Nabs), the oncolytic virus was incubated with a fluorescently Cy5-labeled aptamer. After incubating for 1 h, serum with Nabs was added and the samples were analyzed within 24 h (Figure 2).

The aptamer NV14t_56 remained stable for 1 h in the presence of serum. However, it was subsequently hydrolyzed by serum nucleases. The stability of the aptamer over a one-hour period was also confirmed by electrophoretic analysis (Appendix A, Appendix A). These findings align with literature data indicating that aptamers can resist nuclease degradation for several tens of minutes, or even longer with appropriate modifications [22].

The dissociation constant was determined according to a previously published protocol for aptamers [15]: the oncolytic virus VV-GMCSF-Lact (10^7^ PFU) was incubated with different concentrations of the aptamer Cy5- NV14t_56 (5 nM, 50 nM,100 nM, 200 nM, 500 nM). Using the mean fluorescence intensity (MFI) of the samples (virus–aptamer complex), we plotted it against different concentrations of the Cy5-labeled aptamer (5–500 nM) and then analyzed it using nonlinear regression analysis (Figure 3). 

The binding assay showed that the aptamer NV14t_56 bound with medium affinity to VV-GMCSF-Lact in the micromolar range (Bmax = 46,014, Kd = 352.8 nM). 

### 2.3. Determination of the Size of the Virus–Aptamer Complex and ζ-Potential, the Effect of Human Blood Serum Containing Neutralizating Antibodies on the Virus–Aptamer Complex

To determine the effect of the aptamer NV14t_56 on viral particle aggregation, dynamic laser light scattering was used. After 30 min of co-incubation, the following characteristics were determined: size by intensity: 556 ± 9 nm (97–99%), 5451 ± 54 nm (1–3%); size by number: 505 ± 11 nm; z-average: 561 ± 9 nm; PDI: 0.35 ± 0.02. ζ-potential: −7.9 ± 0.8 mV. The size by intensity and the size by number are comparable to the size of a viral particle. Determination of large aggregates by intensity (size by intensity: 5451 ± 54 nm (1–3%)) indicates the presence of single aggregates in small quantities. The interaction of the virus itself with the aptamer was also analyzed over a time period of 1 h 40 min after joint incubation (Figure 4). The increase in the size and intensity of conjugates in the time range of 1–5 min may be associated with conformational rearrangements during the interaction of the virus and the aptamer and the establishment of equilibrium in the solution.

The addition of the aptamer NV14t_56 to the virus caused a decrease in the value of the hydrodynamic diameter from 805 ± 15 nm to 556 ± 9 nm (by intensity) (Table 1; Samples 1 and 3). This is possibly an indication of aptamers enveloping viral particles and breaking up any aggregates; however, further investigation and confirmation by other methods, such as transmission electron microscopy, is needed to draw any conclusions. 

Serum dilutions of 1:500, 1:1000, 1:3000 (Samples 4–6, respectively) did not affect the Z-average values and did not statistically significantly differ from each other. The ζ-potential of the VV-GMCSF-Lact is −8.4 ± 0.4 mV (Sample 1) and −14.6 ± 0.4 mV in the sample with added aptamers (Sample 3), which indicates an increase in the stability of the complex. It is worth noting that it was possible to select a suitable model for calculating spherical particles in solution only for samples 5 and 6. In sample 4, the higher concentration of serum components probably prevented reliable values from being obtained.

### 2.4. Using Microscale Thermophoresis (MST) for the Characterization of the Interaction of Aptamers with VV-GMCSF-Lact and Human Blood Serum Containing Neutralizing Antibodies

We used the MST method to evaluate molecular interactions between aptamer and virus. By varying the titer of the virus (VV-GMCSF-Lact) at a constant aptamer concentration, we observed a significant inflection on the binding curve, likely indicating the binding process (Figure 5A). Using the Hill equation to fit the data, we determined the value of the half-maximal effective concentration EC50 = 1.487 × 10^9^ PFU/mL (adjusted R² = 0.9966).

Next, we evaluated the stability of the aptamer–virus complex in the presence of serum containing NAbs. Due to the intricate composition of the serum, our objective was to qualitatively assess the disintegration of the complex upon interaction with serum components. To achieve this, we analyzed the ΔF_norm_ parameter for the aptamer, aptamer–virus complex, aptamer–serum mixture, and aptamer–virus complex in the presence of different serum dilutions. 

The ΔF_norm_ parameter exhibits an inverse dependence on the MST amplitude, which decreases with the increase in of the substrate bound fraction. The lowest ΔF_norm_ value was expectedly observed for the unbound aptamer. In the presence of viral particles (1.0215 × 10^9^ PFU/mL) and 10x serum dilution, ΔF_norm_ increased 1.043-fold and 1.034-fold, respectively. The complete mixture comprising the aptamer–virus complex with 10× diluted serum showed the highest ΔF_norm_ value, which was 1.052-fold greater than that of the unbound aptamer. The effect of serum on ΔF_norm_ values persisted up to approximately 150× dilution, while ΔF_norm_ values did not fall below the typical values of the aptamer–virus complex (Figure 5B,C). In summary, these observations suggest that serum components likely bind to the aptamer–virus complex without disrupting it, as the released labeled aptamer would presumably reduce ΔF_norm_ even below the typical values of the aptamer–virus complex.

### 2.5. Cytotoxic Effect of Oncolytic Virus VV-GMCSF-Lact on U87 MG Cells in the Presence of the Aptamer NV14t_56 and Human Blood Serum

To determine the synergistic effect of the aptamer NV14t_56 and the antagonistic effect of the human blood serum with NAbs against the oncolytic virus VV-GMCSF-Lact, we conducted the MTT (Figure 6). Two serum samples, one containing neutralizing antibodies against vaccinia virus and one without such antibodies, were used along with a 200 nM concentration of the NV14t_56 aptamer. The positive serum sample exhibited effective neutralization of the virus, as determined by the 50% Plaque Reduction Neutralization Test (128 PRNT50/mL). 

Previously, it was shown that compared to immortalized cell cultures, as well as primary cultures, U87 MG is one of the most resistant to VV-GMCSF-Lact [23]. In accordance with the results of the previous study, the cytotoxic dose of VV-GMCSF-Lact equal to IC50 = 0.1 PFU/cell was used for this experiment. We deliberately used the minimum dilution of the serum possible for this experiment (1:10) to achieve the maximum neutralizing effect. Figure 6 shows that the virus is effective against U87 MG tumor cells in the presence of fetal bovine serum (A + FBS vs. V + FBS or V + A + FBS (*p* ≤ 0.0001)).

When serum with a high content of neutralizing antibodies (S1) was used, the virus’ effect was blocked, and there was no decrease in tumor cell viability. No significant differences were observed between the groups (A + S1, V + S1, V + A + S1). In contrast, when serum without neutralizing antibodies (S2) was added, there was a sharp decrease in tumor cell viability, and significant differences were observed between the groups (A + S2 vs. V + S2 (*p* ≤ 0.001), A + S2 vs. V + A + S2 (*p* ≤ 0.0001)). Although the optical density (OD) for the V + S2 group (0.6011) differed from that of the V + A + S2 group (0.5204), indicating higher viability for the V + S2 group, there was no statistical difference between them.

## 3. Discussion

Virotherapy is a promising approach to cancer treatment that that combines the principles of immunotherapy by mobilizing the body’s natural defenses and directly lysing infected tumor cells. When the virus enters the body, it triggers the production of virus-neutralizing antibodies, which can reduce its antitumor effect. To address this issue, various approaches have been proposed, including antiviral immunosuppression; selection of a viral vector used to create an oncolytic virus; a choice of approaches for administering a viral drug; and shielding. The latter proposal involves using this method to enhance the antitumor effectiveness of a recombinant virus. Aptamers are suggested as a means to protect the viral envelope from the immune system’s impact. Various aptamers have been developed to reduce virus aggregation, stabilize the virus at lower temperatures, increase its infectivity, and enhance its stability in human blood serum. 

Aptamers are DNA or RNA containing oligonucleotides that specifically bind to their molecular or cellular ligand [9]. The sequence of aptamers that specifically bind to their target can be determined using the Systematic Evolution of Ligands by Exponential Enrichment (SELEX) method. In this work, an attempt was made to evaluate the nature of the interaction of the virus–aptamer complex with serum containing neutralizing antibodies, as well as its cytotoxic effectiveness against human glioblastoma U87MG cells.

We used previously selected aptamers that specifically bind to live vaccinia virus (VACV, Jennerex Inc., Ottawa, Canada), which made it possible to distinguish between viable and nonviable viruses and were integrated into impedimetric aptasensors [14]. Molecular dynamics simulations were employed to determine the secondary and tertiary structure of the aptamer and its probable binding site (Figure 1). Aptamer truncation is a highly sought-after procedure due to its advantages, such as ease of chemical modification and synthesis, as well as cost-effectiveness. Typically, the aptamer’s interaction with the target involves 10–15 nucleotides, which are observed in various structural elements, such as hairpin loops, G-quartet loops, bulges, or pseudoknots [24,25]. The truncated NV14t_56 aptamer holds promise for further research as part of a new bifunctional aptamer. Synthesizing long aptamers (over 90 nt) poses technical challenges. Therefore, for the future bivalent aptamer, a shorter NV14t_56 aptamer is preferred. In the case of the NV14 aptamer, we opted to remove the primer nucleotides at both the 5’ and 3’ ends. This allowed us to eliminate more than twenty nucleotides while preserving important structural features, such as the two hairpin loops. Moreover, the removed nucleotides are unlikely to contribute to specific binding as they mainly form a double helix in this context. It is worth noting that in the future, it would be better to use the native aptamer (NV14t) as a control rather than simply comparing truncated aptamers to each other, even if one binds significantly better than the others [21].

Using flow cytometry, we detected a decrease in the average fluorescence value after 1 h of incubation with serum (Figure 2). Indeed, the aptamer is an unstable molecule; when administered intravenously, the aptamer quickly begins to hydrolyze under the action of nucleases contained in the serum. To increase the stability of the aptamer, it is either modified or conjugated with various composite structures (liposomes, dendrimers, nanoparticles, etc.) [26]. However, when using them in situ, this time is quite sufficient for diagnosing tumor cells [27]. The binding assay (Figure 3) showed that the aptamer NV14t_56 bound with medium affinity to VV-GMCSF-Lact in the micromolar range (Kd = 0.3528 μM). 

Since the virus enters the blood, lymph (if we consider the macro level), and cells (micro level), we were interested in determining whether it is capable of forming aggregates in the presence of the aptamer and serum. Dynamic light scattering (DLS) is a powerful tool used to study the diffusion behavior of macromolecules in solution, allowing one to determine hydrodynamic radii and charge of the resulting aggregates [28]. The DLS data indicated that the aptamer–virus mixture was generally homogeneous in size, intensity, and quantity. These characteristics were comparable to the size of the viral particle (Figure 4). Based on the size-by-intensity parameter, the solution was nearly monodispersed, with aggregates only making up 1–3% of the particles.

Based on DLS data, it can be assumed that aptamers envelop viral particles and prevent the formation of any aggregates. Serum dilutions did not affect aggregation or the Z-average values (Table 1). The ζ-potential is an important indicator of the surface charge of particles and a measure of electrostatic interaction (repulsion or attraction) between particles, as well as one of the main parameters influencing the stability of disperse systems [29]. The magnitude of the ζ-potential is predictive of the colloidal stability of the solution [30]. Both values (−8.4 ± 0.4 mV (Sample 1) and −14.6 ± 0.4 mV in the sample with added aptamers (Sample 3) indicate an increase in the stability of the virus–aptamer complex. 

Microscale thermophoresis (MST) is a biophysical technique used to quantify interactions between molecules such as proteins and small molecules. The MST assay is used to detect protein–protein and protein–drug interactions by quantifying the thermophoretic movement of fluorescent molecules in response to a temperature gradient. In this study, we used MST as an alternative method to detect the formation of aptamer–virus complex. Indeed, the normalized fluorescence curve had a clear inflection that most likely describes the binding process of the aptamer to viral particles with EC50 = 1.487 × 10^9^ PFU/mL (Figure 5A). Furthermore, analysis of amplitudes of MST curves (for the pure aptamer, aptamer–serum mixture, aptamer–virus complex and the aptamer–virus complex in the presence serum dilutions) reveal the binding affinity of serum components to the aptamer–virus complex (Figure 5B). The observed binding effect was meaningful up to a 150-fold dilution of serum without any evidence of aptamer–virus complex disruption (Figure 5C). Despite the fact that the virus has several binding sites for the aptamer, we obtained the standard dose–response curve, which is sometimes called the four-parameter logistic equation. It fits the bottom and top plateaus of the curve, the response data (EC50), and the slope factor (Hill slope) [31]. Parameterization of occupancy (binding) is achieved using the Kd parameter, and shows how many aptamers we need to reach this half-maximal effect [32]. 

We also investigated the interaction of the virus VV-GMCSF-Lact with the aptamer NV14t_56 in the presence of human blood serum with or without Nabs (Figure 6). We conducted a standard test for the determination of cytotoxicity of the oncolytic virus VV-GMCSF-Lact against human glioblastoma cells U87 MG. In the experiment with negative serum without Nabs (S2), we see a slight synergistic effect of aptamer and virus, although we can only speculate at the trend level; no statistical significance was obtained. In the presence of serum containing neutralizing antibodies (S1), the oncolytic virus exhibits no no cytotoxic effect. The mechanisms involved in preventing the oncolytic action of the virus when exposed to serum with Nabs can be different. Regarding the neutralization of the antibodies, this may involve binding to or near the viral receptor binding site; preventing attachment by steric obstruction, disassembly or a change in conformation of virus surface entry proteins; prevention of penetration after attachment and action after penetration [33]; but not virion aggregation, because we did not see this from the dynamic light scattering data. The lack of immunoshielding of the aptamer here may be due to the fact that we have not achieved optimal concentration conditions, or that there is variability in the antigenic structure of surface proteins of different VACV strains used to construct recombinant variants—in the original article, Wyeth VACV strain (recombinant JX-594) [14], but in ours, LIVP VACV strain (recombinant VV-GMCSF-Lact). Accordingly, although the aptamer binds to the virus, it does not completely protect viral epitopes from the action of neutralizing antibodies. To overcome this problem, it may be useful to either evaluate the remaining truncated aptamers for immunoshielding [21] or perform a SELEX procedure on this particular virus strain, VV-GMCSF-Lact. Additionally, to overcome this issue, one of the possible solutions is to enhance the virus targeting by using a bifunctional aptamer that can bind both to the virus and to neutralizing antibodies [26].

Thus, in this study, we characterized the interaction between an aptamer and an oncolytic virus in the presence of serum with and without neutralizing antibodies for the first time. We determined the dimensions of the conjugate, its charge, dissociation coefficients, and the efficiency of the connection. When using an aptamer to shield a virus from neutralizing antibodies, it is important to consider the level of neutralizing antibodies in the patient’s blood. 

## 4. Materials and Methods

### 4.1. Molecular Modeling

The mFold [34] program was used to predict the secondary structures of the aptamers, considering experimental parameters such as the folding temperature and the presence of ions in the solution. The tertiary structures of the aptamers were modeled using the SimRNA [35] and VMD [36] programs. For the molecular dynamic simulations, the GROMACS 2019.8 [37] package was used. The simulations were carried out for a duration of 200 nanoseconds (ns). To describe the interactions between atoms and water, the Amber14sb [38] force field and the TIP3P water model [39] were used. The aptamer was solvated in a periodic cubic box of water. The negative charge of the aptamers was neutralized with Na+ ions. Additional Na^+^ and Cl^–^ ions were placed in the system to reach the concentration 0.15 M. The MD simulations were performed in the NPT ensemble, maintaining a constant number of particles (N), pressure (P), and temperature (T). The temperature was set to 310 K, and the pressure was maintained at 1 atmosphere (atm) using the velocity-rescaling thermostat [40] and the Parrinello–Rahman barostat [41]. The clustering analysis of the obtained trajectories was performed using the quality threshold algorithm implemented in the VMD program [42].

### 4.2. Oligonucleotides Sequence

The sequences of the aptamers to the vaccinia virus envelope were taken from [14]. Further, using mathematical modeling, they were shortened (up to 56 nucleotides) and synthesized in Lumiprobe (Moscow, Russia). Thus, the primary sequences of aptamers 8NV37_14 (variant 2_12-67) used in the study were [Cy5]-GTAACCAGCCATCACCCTATTATCTCATTATCTCGTTTTCCCTATGCGGCATAGG. Aptamers were pre-renatured before use. To achieve this, fluorescently labeled DNA aptamers were heated to 95 °C for 5 min, ice-cooled for 2 min, and incubated at 37 °C for 15 min.

### 4.3. Cell Lines

The cancer cell line U-87 MG was sourced from the Russian cell culture collection (Russian Branch of the ETCS, St. Petersburg, Russia). U-87 MG cells were grown in α-MEM with nucleosides (Gibco, Waltham, MA, USA), supplemented with 10% of fetal bovine serum (Gibco, Waltham, MA, USA), 1 mM L-glutamine, and 1% (*v*/*v*) antibiotic–antimycotic solution (Gibco, Waltham, MA, USA). Cells were cultivated in 25 cm^3^ tissue culture flasks in a humidified 37 °C incubator supplied with 5% CO_2_ and were passaged with the TripLE Express Enzyme (Thermo Fisher Scientific, Waltham, MA, USA) every 3–4 days.

### 4.4. Serum Samples

Two samples of human blood serum were used in our work: one from a person vaccinated with the smallpox vaccine (Lister strain) and containing anti-vaccinia neutralizing antibodies (S1); the second was from a naive donor who had never received a small-pox vaccine (S2). The titer of virus neutralizing antibodies (NAbs) was determined by the method described in [43]. The neutralization efficiency of S1 serum was calculated relative to the number of plaques with non-immune serum (S2), as the highest S1 serum dilution, at which 50% neutralization of the virus was achieved. The titer of NAbs was expressed as a 50% plaque-reducing neutralizing titer in 1 mL of serum (PRNT50/mL). The vaccinia virus-neutralizing antibody titer in S1 serum was 128 PRNT50/mL.

### 4.5. Flow Cytometry

To determine the interaction between various aptamers and neutralizing antibodies, flow cytometry (FC) was carried out using a FACSCantoII flow cytometer (BD Biosciences: Franklin Lakes, NJ, USA), and the data were analyzed by FACSDiva Software v9.0 (BD Biosciences, Franklin Lakes, NJ, USA). First, solutions containing 10^7^ PFU of VV-GMCSF-Lact particles and 0.1 mg/mL of masking yeast RNA were pre-incubated at 25 °C for 30 min. The renatured aptamers NV14t_56 were added to the viral particle solutions to achieve a final concentration of 200 nM and incubated at 25 °C for 30 min. After that, anti-VV serum was added to the samples in different ratios (1:3000, 1:1000, 1:500, 1:100) and incubated for 60 min at 37 °C, with subsequent resuspension in PBS and measurement on the cytometer. For a serum dilution of 1:500, measurements were taken at many time points (0 min, 10 min, 30 min, 1 h, 4 h, 7 h, and 24 h). To calculate the equilibrium dissociation constant (Kd) of the interaction of aptamers and VV-GMCSF-Lact (with constant concentration of 10^7^ PFU), the average MFI value from at least three independent experimental replicates was plotted against aptamer concentration (5 nM, 50 nM, 100 nM, 200 nM, 500 nM) and curved using nonlinear regression analysis (Graphpad Prism 6, Graphpad Software, Inc., La Jolla, CA, USA). The Kd value was obtained using the one site-specific binding equation: 

Aptamer_bound_ = (B_max_ × C_aptamer_): (K_d_ + C_aptamer_), where: B_max_ = maximum binding sites; C_aptamer_ = concentration of aptamer; K_d_ = dissociation constant (binding affinity) [15].

### 4.6. Dynamic Light Scattering

The dynamic light-scattering (DLS), zeta-potential and polydispersity index measurements were taken on a Malvern Zetasizer Nano device (Malvern Instruments, Worcestershire, UK) at 37 °C in solutions with different ratios of viral particles (2. 8 × 10^8^ PFU/mL) to aptamer (10 μM) and different dilutions in serum [44]. All DLS results were calculated as the average of at least triplicate measurements and presented as mean ± SD. 

### 4.7. Microscale Thermophoresis

The binding of the aptamer NV14t_56 to VV-GMCSF-Lact viral particles was measured using microscale thermophoresis [45]. Each mixture contained 0.05% Tween 20 to minimize any non-specific interactions and sample aggregation. First, Amicon Ultra-0.5 Centrifugal Filter Unit (UFC5010BK, Merck KGaA, Darmstadt, Germany) was used according to the manufacturer’s instructions to remove debris from the sample and increase viral particle concentration to the 109 PFU/mL range. Then, multiple concentrations of the viral particles were created by performing serial dilutions in 1mM Tris. The aptamer NV14t_56 was then added to each sample to achieve a final concentration of 20 nM. Measurements were carried out using standard capillaries in the Monolith NT.115 device (NanoTemper Technologies GmbH, München, Germany). All calculations were based on three independent experiments using GraphPad Prism 6.01. We used a standard dose–response curve, which is called a four-parameter logistic equation, to determine the lower and upper plateau of the curve, EC50 and slope coefficient (Hill slope). 

### 4.8. Cytotoxicity Analysis

To determine the effect of the combined use of aptamers and VV-GMCSF-Lact on cell proliferation of U-87 MG glioblastoma, the cell culture MTT test was used. U-87 MG cells were seeded in 96-well plates at a density of 6·10^3^ cells per well. After 72 h of incubation, they were washed with PBS once and incubated with different combinations of VV-GMCSF-Lact, the aptamer NV14t_56 and serum (dilution 1:10) for another 72 h. After that, the MTT assay was performed. Optical density was recorded using a microplate reader (Apollo-8 Microplate Absorbance Reader LB 912, Berthold Technologies, Bad Wildbad, Germany) at 570 nm, with a reference wavelength of 620 nm. 

### 4.9. Statistical Analyses

Outcome variables are expressed as means ± standard deviations (SD), and each experiment was repeated at least three times. Statistical analyses were performed using GraphPad Prism 6.01 (GraphPad Software, San Diego, CA, USA). To compare more than two data sets, we used two-way analysis of variance. Differences were considered significant if the *p* value was <0.05.

## 5. Conclusions

In this work, mathematical modeling was utilized to characterize the secondary structure of an aptamer, and demonstrated the most effective binding to the oncolytic virus VV-GMCSF-Lact in preliminary cytometric experiments. For the first time, the molecular interactions between the aptamer and the VV-GMCSF-Lact virus and the cytotoxicity of the virus when combined with the aptamer in the presence and absence of neutralizing antibodies were assessed. Based on DST data, it was shown that the virus with an aptamer is a relatively homogeneous solution, indicating that the aptamers likely envelop the viral particles and prevent aggregate formation. The obtained binding parameters (Kd and EC50) of aptamers with the VV-GMCSF-Lact virus in the presence of serum are the initial important steps for a better understanding of the mechanism of interaction of aptamers with VV-GMCSF-Lact and for evaluating their potential application in virotherapy.

## Figures and Tables

**Figure 1 molecules-29-00848-f001:**
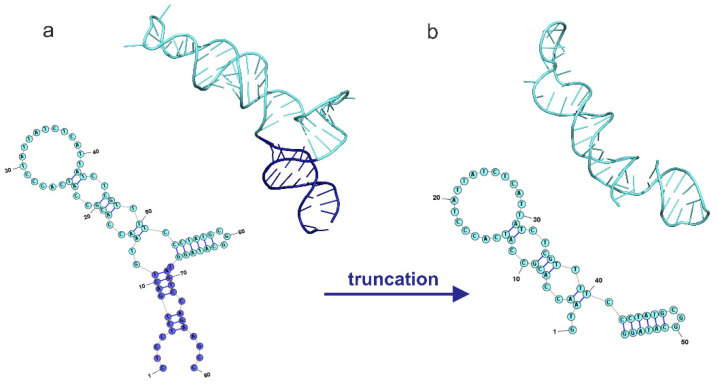
Secondary and tertiary structures of the initial NV14 (**a**) and truncated NV14t_56 (**b**) aptamers. Aptamer NV14 was truncated by 11 nucleotides from the 5′ end and 12 nucleotides from the 3′ end (colored in dark blue).

**Figure 2 molecules-29-00848-f002:**
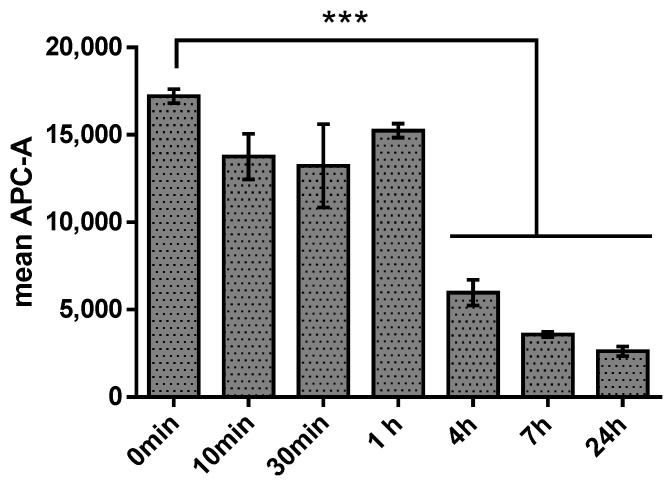
Cytometric analysis of the stability of the aptamer–virus complex in the presence of human blood serum with NAbs. The fluorescence signal for the Cy5 dye was detected in the APC-A channel. Three asterisks (***) indicate *p* ≤ 0.001.

**Figure 3 molecules-29-00848-f003:**
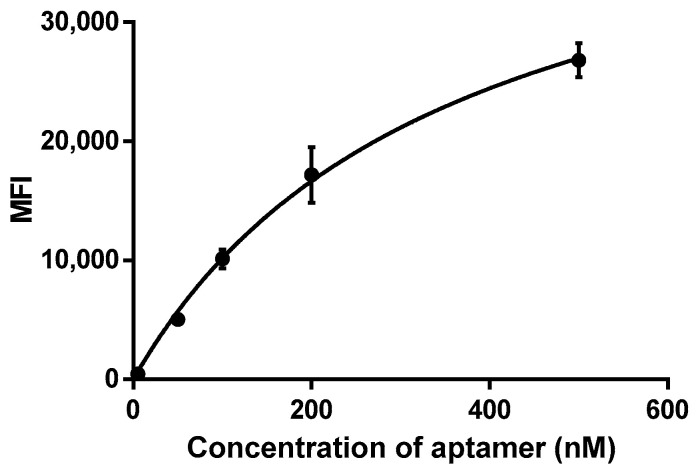
Determination of binding affinity, Kd value of the aptamer NV14t_56 by flow cytometry. The binding affinity of NV14t_56 to VV-GMCSF-Lact was determined by flow cytometry using Cy5-NV14t_56. The mean fluorescence intensity (MFI) of various concentrations (nM) of Cy5-NV14t_56 was plotted to determine the dissociation constant, Kd.

**Figure 4 molecules-29-00848-f004:**
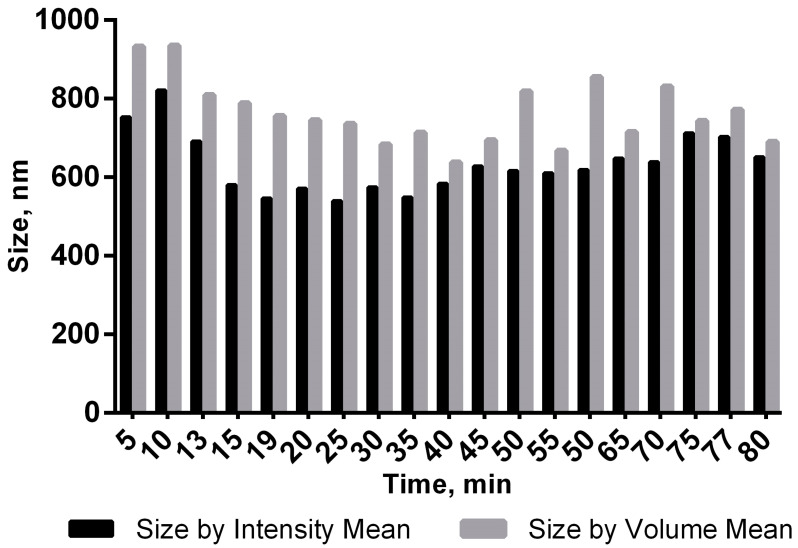
Graph of the size of the aptamer–virus complex versus time.

**Figure 5 molecules-29-00848-f005:**
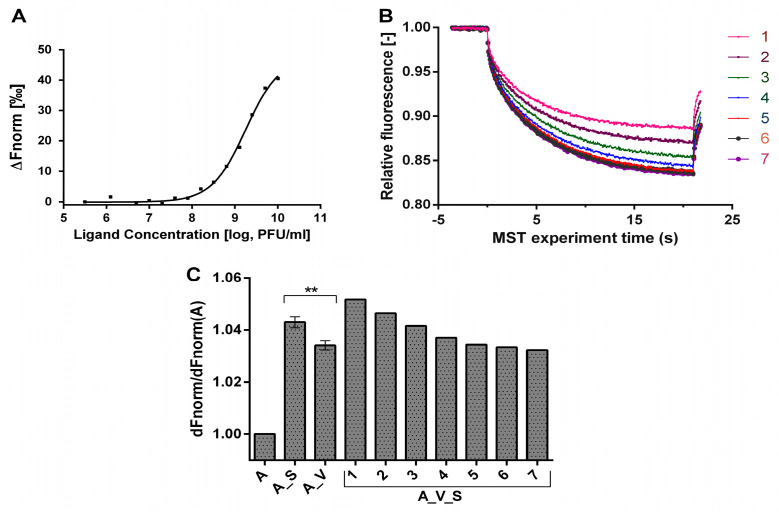
(**A**) Dose–response curves for the binding interaction between aptamer and VV-GMCSF-Lact; (**B**) MST-Trace Chart of the serial dilution of serum, starting from “1:10” (1) with a dilution factor of 2.5 to “1:2441.4” (7); (**C**) Normalized fluorescence of samples as a function of serum dilution. The ratios of ΔFnorm of each sample to ΔFnorm of pure aptamer are plotted. Labels: NV14t_56 aptamer (A), VV-GMCSF-Lact (V), human blood serum with Nabs (S). Two asterisks (**) indicate *p* ≤ 0.01.

**Figure 6 molecules-29-00848-f006:**
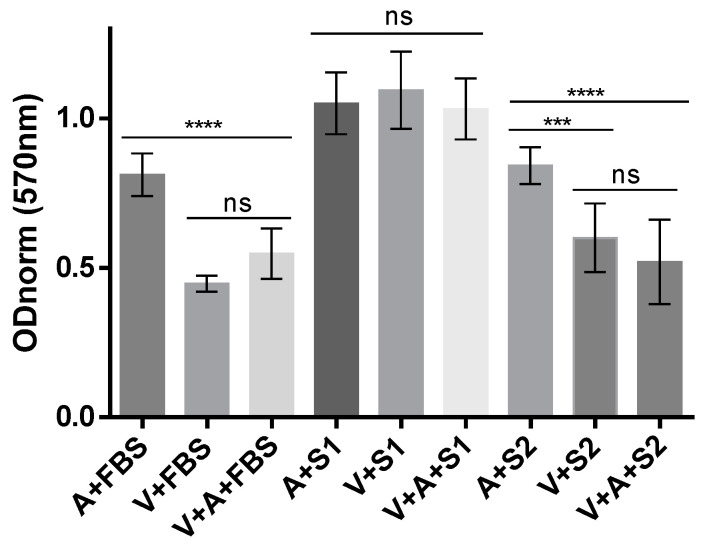
Cytotoxic effect of the oncolytic virus VV-GMCSF-Lact (V) with the addition of aptamers (A) and various serums: fetal bovine serum (FBS), serum with NAbs (S1), serum without NAbs (S2). Three asterisks (***) indicate *p* ≤ 0.001. Four asterisks (****) indicate *p* ≤ 0.0001. The abbreviation “ns” indicates non-significance, *p* > 0.05.

**Table 1 molecules-29-00848-t001:** DLS data for the VV-GMCSF-Lact virus after incubation with the aptamer NV14t_56 in the presence of various amounts of serum.

		Hydrodynamic Diameter (by Intensity), nm	ζ-Potential, mV
1	Virus (20:80 dilution)	805 ± 15	−8.4 ± 0.4
2	Aptamer_Cy5 (20:80 dilution)	4.3 ± 0.7	
3	Virus + Aptamer_Cy5 (10:10:80)	556 ± 9	−14.6 ± 0.4
4	Virus + Aptamer_Cy5 (1:1 ratio, serum dilution 1:500)	571 ± 30	
5	Virus + Aptamer_Cy5 (1:1 ratio, serum dilution 1:1000)	542 ± 14	
6	Virus + Aptamer_Cy5 (1:1 ratio, serum dilution 1:3000)	512 ± 24	

Note: Samples were diluted in buffer, as indicated in the table.

## Data Availability

Data are contained within the article.

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
