# Peer review of "Characterizing Aptamer Interaction with the Oncolytic Virus VV-GMCSF-Lact"

_molecules, 2024, doi:10.3390/molecules29040848_

Round 1

Reviewer 1 Report

Comments and Suggestions for Authors

It's a very good piece of work. However, I just have few queries. 

1. Please write the parameters of truncation.

2. In Introduction, explain how this work is different from previously published work.

3. Figure 3, Y-axis abbreviation must be expanded. Same issue with Figure 2. 

4. In table 1, why zeta potential values were not mentioned for all?

5. Figure 5 needs to be rearranged to be symmetry.

Comments on the Quality of English Language

Minor improvements needed.

Reviewer 2 Report

Comments and Suggestions for Authors

The manuscript by Dymova et el. ‘Characterizing aptamer interaction with the oncolytic virus 2 VV-GMCSF-Lact’ describes the effect of aptamer to oncolytic virus on virus aggregation and oncolytic efficiency. I do not recommend this manuscript for the publication in ‘Molecules’ as conclusions are not supported with results. Scientific novelty is rather low.

Major comments.

1.       Abstract states that ‘The purpose of this study was to characterize the aptamer previously selected for VACV, evaluate the aggregation and molecular interaction of the optimized aptamers with the recombinant oncolytic virus VV-GMCSF-Lact, and estimate their immunoshielding properties in the presence of human blood serum’. The manuscript shows week binding to viruses with micromolar Kd, no obvious effect on the aggregation, no effect on oncolytic activity, no immunoshielding. Thus, these results are of low interest for the reader.

2.       The manuscript states that a truncated aptamer is characterized. However, there are no comparison with parent aptamer. The refs contain no data on truncated aptamer NV14t_56. So, it is unclear, whether the aptamer provides some improvements compared to previous works.

3.       The manuscript states that ‘The resulting aptamer may exhibit comparable or greater binding affinity to the target compared to the initial NV_14 aptamer. This is due to the fact that the duplex part … can create steric hindrances for a tighter interaction be111 tween the NV14 aptamer and the target.’ The results do not contain any clues on the improvement or even retention of the initial affinity. So, this statement is not supported with the data. There are a lot examples on the end-duplex involvement in target binding in the field, so this point can be wrong.

4.       ‘Similarly, we truncated all aptamers proposed in [16], but after a series of preliminary experiments using flow cytometry which are described in [21], we determined that NV14t_56 bound most efficiently to our oncolytic virus, VV-GMCSF-Lact, so in subsequent experiments we only used this aptamer.’ The refs contain no data on NV14t_56.

5.       Subsection ‘Stability of aptamer -virus comlex in the presence of human blood serum and aptamer binding to VV-GMCSF-Lact’ describes the decrease of aptamer-viral complex stability in the presence of human serum. The interpretation of these data is unclear. The destabilization can be due to off-target interactions as well as due to neutralizing Ab or nucleases. The mechanism is to be described using additional techniques that are common in the field. E.g. electrophoresis to study nuclease activity, comparison of the complex stability with and without serum, with and without neutralizing Ab.

6.       Next subsection states that ‘After 30 minutes of co-incubation … the size by intensity and the size by number is comparable to the size of a viral particle.’ However, we can see the increase in virus size during the incubation instead of expected decrease. So, aptamer provided the opposite effect. The table 1 is very confusing, as here we can see the size of aptamer about 200 nm (it is to be 100 times lower) along with very large viruses without aptamers that are much larger than starting point in kinetic experiment. DLS is bad technique for polydisperse samples. Some other approach is necessary to exclude the obvious artefacts. The charge of the virions was not increased in the presence of the aptamers (-10 vs. -7.6). Possibly, the assembly of the complex was not achieved at all.

7.       MST shows some non-specific interaction with serum. It is a well-known problem with MST. I not catch, whether the detergent was used in this experiment as materials and methods are not detailed. Detergents are necessary for MST to exclude non-specific interactions and sample aggregation.

8.       Subsection 2.5. Aptamer does not provide any advantages for the oncolytic therapy. No synergetic effects were shown. This part was interpreted in a wrong way.

Discussion. ‘Thus, in this work, the nature of the interaction of an aptamer with an oncolytic virus in the presence of serum with and without neutralizing antibodies was characterized for the first time’ No, there is no data on the nature of the interaction with and without neutralizing antibodies. ‘The dimensions of the conjugate, its charge, dissociation coefficients and the efficiency of the connection were determined.’ No, these data are rather questionable. They are to be rechecked with other techniques. ‘When using an aptamer to shield a virus from neutralizing antibodies, the level of neutralizing antibodies in the patient's blood should be considered’. No, there is no data on immunoshielding. The aptamer was not proved to be a competitor to Ab.

Minor comments.

1.       English is to be checked by a native speaker

2.       What is glioma dynamic light scattering in the keywords?

3.       VACV and other abbreviations are to be explained at the first use

4.       Abstract: Self-evident sentences are to be omitted. For example, ‘Biophysical methods can be useful for selecting and studying the interaction of viruses and ligands of any nature, including DNA aptamers, which is one of the first necessary steps to assess their applicability in virotherapy.

Comments on the Quality of English Language

The language is to corrected

Reviewer 3 Report

Comments and Suggestions for Authors

The manuscript titled “Characterizing aptamer interaction with the oncolytic virus VV-GMCSF-Lact” by Dymova, investigated aptamers’ potential to enhance virotherapy. The aptamer for VACV remained stable inhuman blood serum, bound effectively to oncolytic virus VV-GMCSF-Lact, and prevented viral particle aggregation. In vitro experiments demonstrated the aptamer’s effectiveness in preventing virus aggregation in the presence of human blood serum without neutralizing antibodies. Aptamers may impact oncolytic storage and usage efficiency, making them promising for virotherapy. This is a novel aspect in a field of intense research. Still some issues need to be clarified, as listed below, before the manuscript can be accepted for publication in molecules.

1.      The choice of test conditions in Line 94 is not explicitly mentioned, and additional information regarding the rationale behind this selection is needed. I recommend providing references to support the basis for the chosen testing conditions. This will enhance the transparency of the experimental design and help readers understand the scientific justification for the specific conditions employed in the study.

2.      Before discussing the experimental results, it would be helpful to provide a brief introduction to the naming convention of the NV14 aptamer mentioned in line 101. Specifically, clarify whether NV14 is the aptamer selected through SELEX, and if so, briefly explain the basis for its designation. This information will provide context for readers and improve the understanding of the aptamer’s origin and relevance in the study.

3.      The manuscript mentions that the truncated aptamer exhibits improved binding affinity. Before delving into the experimental results, it is essential to discuss why such truncation enhances binding capability. Although reference 21 is cited for corresponding truncated aptamer data, please provide a more detailed explanation in the manuscript.

4.      In the sentence on P9, L268-L271, the authors discuss the advantages of modification or linkage to enhance aptamer stability and allow sufficient time for tumor cell diagnosis. However, the choice of aptamer truncation is not clearly justified. Given that modifications or structural connections offer benefits, the authors should explain the rationale behind selecting aptamer truncation. Furthermore, the two clauses in this sentence do not establish a contrasting relationship. It is recommended to provide clarification and rephrase the sentence for improved coherence.

5.      In the Discussion section, it would be beneficial to explicitly specify which figure from the results is being discussed. This will provide clarity for readers and help them connect the discussion points with the corresponding visual representation in the manuscript.

6.      In line 312 of page 9, it would be helpful to elaborate on how the changes in viability are observed in the experimental results. Provide a clear explanation or reference to the specific data or figures that demonstrate the variations in viability. This clarification will enhance the understanding of the experimental outcomes related to viability changes.

Round 2

Reviewer 2 Report

Comments and Suggestions for Authors

Dear authors,

Thank you for the point-by-point answers. I understand that the expected aptamer properties were not proven in this case. However, it is necessary to discuss this point honestly. One of the first works in the field demonstarted exciting characteristics of the aptamers (doi:10.1038/mtna.2014.19). So it is very disappointing to see micromolar constants an the absence of immunoshieling ten years later. I suggest to state clearly the absence of immunoshielding in the text and discuss the possible reasons / future directions to overcome this problem. The changes in virus aggregation can be achieved using non-specific cheap molecules; so this part is not so interesting for the reader. I also do not understand why you did not provide a comparison with a parent aptamer; it is a good practice during the aptamer maturation. Also, DLS provided an increase in zetta potential that indicate low level of aptamer binding. Negatively charged aptamers decrease zeta potential. Some technical troubles may cause this effect.
